# In Vitro Investigation of the Effects of Various Reducing Agents on Dentin Treated with Hydrogen Peroxide

**DOI:** 10.3390/polym16111473

**Published:** 2024-05-23

**Authors:** Alaa Alatta, Mohannad Nassar, Mehmet Gorduysus, Walaa Alkhatib, Mahmoud Sayed

**Affiliations:** 1Department of Preventive and Restorative Dentistry, College of Dental Medicine, University of Sharjah, Sharjah P.O. Box 27272, United Arab Emirates; 2Research Institute for Medical and Health Sciences, University of Sharjah, Sharjah P.O. Box 27272, United Arab Emirates; 3Oral Health Science Center, Tokyo Dental College, Tokyo 101-0061, Japan; drmahmoudmacklad@gmail.com

**Keywords:** antioxidant, ascorbic acid, bond strength, dentin, hydrogen peroxide, hardness, *n*-acetylcysteine, reduced glutathione, roughness, sodium thiosulfate

## Abstract

We assessed the effect of non-protein thiols (NPSH), reduced glutathione (GSH) and *n*-acetylcysteine (NAC), on resin shear bond strength (SBS) to hydrogen peroxide (H_2_O_2_)-treated dentin, and their effects on the characteristics of dentin in comparison to ascorbic acid (AA) and sodium thiosulfate (STS). H_2_O_2_-treated dentin was conditioned with 5% AA, GSH, NAC, or STS applied for 1 or 5 min. The positive control group received H_2_O_2_ without antioxidant application, and the first negative control group received distilled water (DW). The specimens received resin bonding immediately after treatment except for the second negative control group (delayed bonding). Microhardness, roughness, and topography were studied. The SBS values of all antioxidants were statistically greater than the positive control group (*p* < 0.05); however, NAC and AA applied for 1 min demonstrated the highest values, which were comparable to delayed bonding. All treatments removed the smear layer except DW, H_2_O_2_, and STS. The negative effect of H_2_O_2_ on resin–dentin bonding was mitigated by the application of the antioxidants; however, their efficiencies were dependent on the antioxidant type and time of application. NAC was more effective in optimizing resin bonding to bleached dentin compared to GSH at 1 min application and STS at both application times but was comparable to AA. Negligible negative effects on the substrate’s roughness and microhardness were detected. The antioxidant properties of the agent and its capacity to remove the smear layer are the processes underpinning the ability of a certain antioxidant to reverse the effect of H_2_O_2_ on bonding.

## 1. Introduction

Due to the increased demand for teeth whitening, the market has seen the introduction of numerous products for this purpose. However, most methods for teeth bleaching utilize hydrogen peroxide (H_2_O_2_) and its precursors [1,2]. This oxidizing agent diffuses into the tooth’s hard tissues, producing unstable free radicals such as hydroxyl and perhydroxyl radicals and perhydroxyl anions that attack double bonds in organic-colored molecules within the tooth structure. Despite the achieved whitening, the process has some negative side effects, including but not limited to changes in tooth substrates and the compromised bonding of resin-based restorative materials to the tooth structure [3,4]. The reduction in the bond strength of adhesive restorations to the tooth structure after tooth bleaching is a major concern in clinical settings, and this is not limited to enamel but also to dentin, particularly in cases where the bleaching material comes into direct contact with dentin, such as the walking bleach technique. The decrease in bond strength is attributed to the presence of active chemicals and highly reactive residual oxygen radicals from the bleaching agent, which interfere with resin bonding, inhibit resin polymerization, and/or increase resin porosity in resin-based restorative materials placed immediately after bleaching [5,6].

Different approaches to enhancing resin bonding to bleached teeth have been investigated; however, the emphasis has now turned towards non-invasive techniques [7]. It has been shown that antioxidants are efficient in hastening the elimination of residual oxygen and free radicals from the tooth structure following bleaching gel application [8]. Antioxidants work by providing free electrons to free radicals or by scavenging or breaking free radical chains to reverse the detrimental effects of bleaching agents. Ascorbic acid (AA) is regarded as the conventional reversal agent due to its ability to restore resin–dentin bond strength to normal levels [9]. AA is a key defense mechanism that accelerates the process of eliminating oxygen from surfaces and, because this substance is biocompatible and water soluble, it can be employed in a number of dental procedures [10,11,12]. Moreover, it is believed that AA has no significant negative effects on the tooth structure; nevertheless, the salt form of AA, sodium ascorbate, has certain disadvantages since it has a yellow hue that may discolor white teeth [13]. Studying resin bonding to bleached tooth structures has received much attention driven by its profound implications in the field of adhesive dentistry. Researchers are currently actively engaged in exploring alternative approaches within this field, driven by the imperative to identify more effective solutions and the focus has been on natural agents over synthetic compounds due to several reasons, which include, but are not limited to, greater biocompatibility and alignment with the recent emphasis on sustainability in healthcare [14]. For instance, recently Macromini et al. recommended the use of 15% alpha-tocopherol to reverse the effect of 40% H_2_O_2_ on bonding to dentin [15]. An ethanol solution, green tea extract, grape seed extract, and curcumin photosensitizer were also recently studied as effective alternatives [16,17].

Sodium thiosulfate (STS) is another antioxidant that is used in the dental field. STS is mainly viewed as the main agent to neutralize sodium hypochlorite (NaOCl) which is the irrigant of choice for root canal treatment procedures. NaOCl has the ability to oxidize dentin, resulting in compromised bonding to root canal and pulp chamber dentin, and STS can attenuate this effect, thus permitting immediate placement of adhesive restorations following endodontic treatment. STS has been favored over AA due to its longer shelf-life [18]. Non-protein thiols (NPSH) are potent antioxidants produced by the human body that help with a variety of biological processes [19]. Reduced glutathione (GSH), the active form of glutathione, is the most common NPSH, accounting for 90% of intracellular levels, whereas *n*-acetylcysteine (NAC) is one of the most bioavailable precursors of GSH [20,21]. The focus on NPSH in dentistry is not new; previous research found that particular NPSH, namely GSH and NAC, were efficient in mitigating the toxic effect of monomers on pulpal cells and inhibiting dentinal MMPs without affecting bonding to dentin [22,23,24]. NAC was suggested to treat oral health issues, such as inflammatory periodontal diseases, because of its ability to efficiently detoxify xenobiotic toxicity, counteract oxidative stress, eliminate invasive microbes, and reduce inflammatory responses [25]. However, no studies using NPSH as reversal agents on H_2_O_2_-treated dentin have been undertaken. If the oxidizing action of bleaching agents is responsible for the decrease in dentin bond strength to composite resin, applying a biocompatible antioxidant prior to bonding may counteract this negative effect [7]. This approach is crucial after the use of bleaching agents, not only to reverse the oxidized status of dentin but also to increase the adhesive bond strength to dentin, or at the very least to bring back the bond strength value to a level comparable to that with unoxidized dentin. Furthermore, as the demand for agents that improve dentin bonding grows, new and improved antioxidant variants are constantly being marketed, and the search for effective and non-toxic compounds continues. Thus, the aims of this study were to evaluate the effect of GSH and NAC on the shear bond strength (SBS) of resin to H_2_O_2_-treated dentin, assess their effects on the roughness, microhardness, and topography of dentin, and compare the results to conventional reversal agents, namely AA and STS. The null hypothesis was that exposure of bleached dentin to the used antioxidants does not affect resin–dentin bonding, or dentin roughness and microhardness.

## 2. Materials and Methods

### 2.1. Specimen Preparation for SBS Testing

The study was approved by the Research Ethics Committee (reference number: REC-21-06-01-01-S) at the University of Sharjah. Seventy-seven sound extracted human molar teeth were collected from the University Dental Hospital Sharjah. The teeth were cleaned of calculus and attached soft tissues and stored in 10% formalin. Flat coronal dentin surfaces were created by sectioning teeth perpendicular to the longitudinal axis of the tooth using a slow-speed diamond saw (Isomet Low Speed Saw; Buehler, Lake Bluff, IL, USA) under water irrigation. The specimens were embedded vertically in a mold filled with self-curing resin (Technovit 4071; Kulzer, Hanau, Germany). A uniform smear layer was created using 600-grit silicon carbide (SiC) paper (Soflex; 3M ESPE, St. Paul, MN, USA) under a water cooling system. The specimens were randomly distributed to 11 groups (n = 7) according to the type of treatment each dentin surface received. The first negative control group (Group 9) received treatment with distilled water (DW) for 15 min, while the positive control group (Group 10) was treated with 35% H_2_O_2_ (Opalescence Endo; Ultradent, South Jordan, UT, USA) for 15 min, followed by rinsing with DW. The experimental groups received treatment with 35% H_2_O_2_ for 15 min followed by rinsing with DW and then conditioning with 5% of either AA, GSH, NAC, STS for 1 or 5 min. All of the specimens were rinsed with DW upon completion of the treatment protocol. The used antioxidant agents were purchased from Sigma-Aldrich. Table 1 summarizes the groups used in this part of the study. The difference between groups 10 and 11 is that the latter received the bonding procedure after 2 weeks and thus is considered as a second negative control group (delayed bonding), meanwhile all other groups received bonding immediately after dentin surface conditioning. The specimens for delayed bonding were stored in 100% humidity at 37 °C.

The bonding agent (Single Bond Universal Adhesive; 3M ESPE, St. Paul, MN, USA) was applied according to the manufacturer’s instructions using a micro-brush with continuous agitation for 10 s, followed by drying with a gentle stream of air for 5 s and was cured for 20 s using an LED light-curing unit (Mini LED; Acteon, Germany). A cylindrical plastic mold with an inner diameter and length of 2 mm was placed on the dentin and secured using a jig (Ultradent, South Jordan, UT, USA). A flowable composite (Filtek Supreme Ultra Flowable; 3M ESPE, St. Paul, MN, USA) was injected into the mold and cured for 40 s. A digital radiometer was used to verify the light intensity of the LED light-curing unit before the polymerization of each specimen. The jig was carefully loosened, and the mold was removed, leaving a cylindrical composite block bonded to the conditioned dentin surface. Specimens were stored in 100% humidity for 24 h for bonding maturation before shear bond strength testing.

### 2.2. Shear Bond Strength Test

A table-top testing machine (SBS tester; Bisco, Schaumburg, IL, USA) was used to test the SBS of the specimens. The specimens were fixed using a specimen holding clamp to hold the bonded dentin surface parallel to the testing machine. Shear forces were applied at the bonded interface with a semicircular metal attachment until failure and running at a crosshead speed of 1.0 mm/min. To calculate the bond strength in megapascal (MPa), the force at the time of failure in newton (N) was recorded and divided by the surface area of the bonded interface.

### 2.3. Specimen Preparation for Microhardness and Roughness Tests

The effect of antioxidants on the microhardness and roughness of H_2_O_2_-treated dentin was evaluated as this would aid in explaining the differences in the results obtained for SBS between the different antioxidants and within the same application time for each antioxidant. Flat coronal dentin discs of 1 mm thickness were prepared from extracted human molar teeth which were further sectioned into 2 sections using a low-speed diamond saw. One section served as a baseline for the microhardness and roughness measurements and the other half received the treatment; the difference in values was taken as the obtained data for Δ microhardness and Δ roughness. After sectioning the discs, the halves were polished using a series of SiC papers of 600 to 2000 grit in an ascending order, under water cooling, followed by a final polishing using one µm diamond paste on a wet grinding wheel. Each disc with its two sections was randomly divided into 8 groups according to the dentin conditioning protocol (n = 7). In each group, the baseline section of each disc was treated with 35% H_2_O_2_ for 15 min, while the other section was treated with 35% H_2_O_2_ for 15 min followed by a 1 or 5 min treatment with 5% of either AA, GSH, NAC, or STS. The sections were rinsed with DW after each agent application. The treatment protocols used in this part of the study are mentioned in Table 2.

### 2.4. Microhardness Measurement

The specimens were mounted on a glass microscope slide with double-sided adhesive tape for fixation and measured using a micro hardness-testing machine (MKV-E hardness tester; Akashi Seisakusho, Kanagawa, Japan). The Vickers diamond indenter was applied to the dentin surface with a load of 50 g for 15 s. Micro hardness was determined at 15 random points on the specimens’ surface. The means of micro hardness measurements for each specimen at each indentation depth were then analyzed. The measurements were converted into a Vickers hardness number (HV) by the monitor, using the equation: HV = 1854 (F/d2); F is the indentation load (g), and d is the diagonal of the indentation (μm).

### 2.5. Surface Roughness Measurement

Scans of the specimen’s surfaces were conducted using a 3D confocal laser scanning microscope (CLSM) (VK-X 150 series; Keyence Corporation, Osaka, Japan) at 10,000× magnification. The obtained data were analyzed using an analyzer program (MultiFileAnalyzer V1.3.1.120; Osaka, Japan). The values of the surface roughness were obtained in the “Sa” parameter.

### 2.6. Smear Layer Removal

To further explain the results obtained by the SBS tests, the effect of each antioxidant on the topography of H_2_O_2_-treated dentin was assessed. The dentin discs were prepared and treated in the same manner as previously described for the microhardness and roughness tests, with the exception of using only SiC paper of 600 grit to create a smear layer and including a group that received only distilled water treatment to compare the negative and positive control groups with the experimental groups in terms of smear layer removal. The specimens were dehydrated with ascending concentrations of ethanol (25%, 50%, and 75% for 20 min, 95% for 30 min, and 100% for 60 min) and then sputter-coated (Quorum Technology; UK, SC7620 mini sputter coater system) with gold/palladium, approximately 10 μm thickness, and observed under a scanning electron microscope (SEM) in 5000× magnification (Tescan Vega 3 xmu; Brno, Czech Republic), running at a 10 kV accelerating voltage.

### 2.7. Statistical Analysis

The SBS, microhardness, and roughness values were estimated using mean values and standard deviations and were checked for significant deviation from normality (Shapiro–Wilk test) and homoscedasticity (Levene’s test). When the normality and equality variance assumptions of the data were valid, two-way ANOVA and Tukey HSD post hoc tests were performed by using the dentin surface conditioning protocol and time of application periods as 2 factors, and then further analyzed by one-way ANOVA and Tukey’s HSD. When the data could not be transformed into a normal distribution, the data were analyzed by a Kruskal–Wallis test, and a Mann–Whitney U multiple comparison test was used to determine specific differences between the groups using SPSS version 22 (IBM Analytics; Armonk, NY, USA). Statistical significance was pre-set at α = 0.05.

## 3. Results

### 3.1. Shear Bond Strength Test

The means and standard deviations of SBS (MPa) are illustrated in Figure 1. The Kruskal–Wallis test revealed significant differences observed between groups (*p* < 0.05). The Mann–Whitney U-test further indicated significantly higher SBS value of delayed bonding protocol compared to all antioxidant groups (*p* < 0.05), except those of AA and NAC applied for 1 min (*p* > 0.05). The delayed bonding protocol also resulted in higher values compared to the positive control group (H_2_O_2_) (*p* < 0.05) and similar results to the negative control group (DW) (*p* > 0.05). Furthermore, all antioxidant groups showed statistically significantly higher SBS values in comparison to the positive control group (H_2_O_2_); however, the lowest increases happened with STS at 1 or 5 min followed by GSH at 1 min. Applying AA or NAC for 1 min showed higher SBS values than the same antioxidant applied for 5 min (*p* < 0.05). However, applying GSH or STS for 5 min resulted in an SBS value that is comparable to applying the same antioxidant for 1 min. Interestingly, NAC and AA applied for 1 min resulted in higher values compared to GSH at the same application time (*p* < 0.05); however, this difference diminished at the 5 min application (*p* > 0.05).

### 3.2. Microhardness

The mean and standard deviation of Δ microhardness are illustrated in Figure 2. The mean Δ microhardnesses for all groups were negative values indicating a decrease in microhardness after using the antioxidants on H_2_O_2_-treated dentin. The Tukey HSD post hoc test indicated significant differences between certain groups. Treatment with GSH for 1 or 5 min resulted in the lowest value of Δ microhardness; however, it was only statistically significant compared to AA applied for 5 min and STS applied for 1 min (*p* < 0.05). NAC and AA at both applications times showed no statistical differences (*p* > 0.05). There were no differences between the same antioxidant applied for 1 or 5 min (*p* > 0.05). Appendix A presents the average microhardness values for the baseline treatment (dentin treated with H_2_O_2_ for 15 min) in each group, along with the microhardness values after treatment of this substrate with the antioxidants for 1 or 5 min.

### 3.3. Surface Roughness

The means and standard deviations of Δ roughness are illustrated in Table 3. Most values were in the positive range, indicating an increase in roughness. However, the 1 min application of GSH or STS resulted in negative values, which indicates a decrease in roughness. The Tukey HSD post hoc test revealed significant differences between certain groups. The highest roughness occurred with GSH applied for 5 min which was statistically different from all other groups (*p* < 0.05) except NAC at the same application time (*p* > 0.05). AA, NAC, and STS applied for 1 min showed similar values to the application of the same antioxidant for 5 min (*p* > 0.05). Appendix A presents the average roughness values for the baseline treatment (dentin treated with H_2_O_2_ for 15 min) in each group, along with the roughness values after treatment of this substrate with the antioxidants for 1 or 5 min.

### 3.4. Smear Layer Removal

SEM images of dentin surfaces with different treatment protocols are shown in Figure 3. Topographical analysis of dentin surfaces after being subjected to DW (Figure 3a) or 35% H_2_O_2_ (Figure 3b) showed the presence of a smear layer; 600 grit SiC paper produced uniform scratches with a smear layer covering the surface of dentin in both groups, this reflects the inability of DW or H_2_O_2_ to remove the smear layer. However, the smear layer was efficiently removed, and the dentinal tubules were exposed after treatment with AA, GSH, or NAC applied 1 or 5 min as shown in Figure 3c–h; respectively. At both application times, STS failed to remove the smear layer (Figure 3i,j).

## 4. Discussion

As the demand for bleaching procedures in dentistry grows, dental clinicians are dealing more with cases in which they need to delay adhesive restorative work for several days post-bleaching, causing needless inconvenience for some patients. In normal circumstances, adhesion to dentin is already a challenging procedure due to the complex structure and variations in morphology, and H_2_O_2_ complicates the matter further. Dentin is the largest component in the tooth structure available for adhesion, and H_2_O_2_ and its associated free radicals have the ability to infiltrate the dentin due to their low molecular weights, and thus we chose dentin as our substrate in the current study [26,27,28]. Several methods have been used in the literature to test the bond strength of resin adhesion to dentin; however, there is no superior method of testing [29]. The shear bond strength method was performed in the current study since it is considered one of the most prevalent in the literature due to its simple sample preparation and lower technique sensitivity compared to other methods [30,31,32]. In this study, we investigated the effect of NPSH on the strength of the bond to dentin treated with H_2_O_2_ and compared it to AA and STS. These agents’ effects on the roughness, microhardness, and topography of dentin were also studied. The findings revealed that H_2_O_2_ has a deleterious effect on dentin adhesion and that delayed bonding restores bond strength to normal levels. Bond strength values comparable to delayed bonding were obtained after using certain antioxidants for a specific time of application. All agents at both application times resulted in negative Δ microhardness values while Δ roughness was dependent on the type of antioxidant and time of application. The created smear layer was successfully removed by AA, GSH, and NAC; however, STS failed to remove it. These results require the rejection of the null hypothesis.

The decrease in the bond strength to the bleached tooth structure is due to the presence of residual peroxides and oxygen on the tooth surface, which interfere with resin bonding by preventing complete polymerization or inducing resin porosities [9,33,34]. For optimal adhesion to a bleached tooth structure, several days must elapse between the last whitening treatment session and adhesive application; however, this delay can be avoided by removing the oxidized enamel/dentin surface layer before bonding, or by applying certain antioxidants after bleaching [35]. We believe that the latter is a promising protocol as it is a more conservative approach. In this study, immediate bonding to dentin treated with H_2_O_2_ without further application of an antioxidant obtained the lowest SBS value. However, delayed bonding restored bonding to normal levels and these results were in agreement with the literature [36]. The inability of H_2_O_2_ to modify the smear layer revealed in our investigation was also confirmed in an earlier report by Karadas et al. [37] where they applied 35% H_2_O_2_ to the dentin surface for 45 min. However, H_2_O_2_ has been reported to modify the dentinal smear layer and expose dentinal tubules after several applications at high concentrations [38]. It is noteworthy to mention that the effect of H_2_O_2_ on adhesion is not solely driven by free radical generation; due to the lower molecular weight of H_2_O_2_, it can easily penetrate the dentin subsurface leading to increased activity of endogenous dentinal MMPs that degrade collagen network, which is essential for the mechanism of resin bonding with dentin [39]. However, this effect would be more enhanced in the long term.

AA is recognized as a main antioxidant and one of the first investigated for its role as a reversal agent against the adverse effects of H_2_O_2_ on resin–dentin bonding [9,37]. A recent study concluded that application of AA salt (sodium ascorbate) solution after internal bleaching with H_2_O_2_ increased resin–dentin bond strength to the same level as unbleached dentin, regardless of the application mode of a multimode adhesive system [37]. In our study, AA resulted in optimal values for bond strength at both application times, which were comparable to the negative control group (DW). The 1 min application of this antioxidant resulted in higher SBS value compared to its 5 min application. The 5% AA employed in this study had a pH of 2.29. Therefore, we hypothesize that the 5 min application may have caused more extensive demineralization of the dentin, impeding optimal resin penetration into the dentinal collagen network, consequently leading to the observed decrease in the SBS compared to the 1 min application. The direct comparison of our results with previous studies is challenging due to different methodologies, concentrations, and application times. Moreover, we used AA while most studies opted for sodium ascorbate for reasons related to the acidic nature of the former. However, as mentioned earlier, sodium ascorbate has its own shortcomings, such as yellow discoloration, thus it seems there is a recent tendency to avoid the application of sodium ascorbate onto the tooth structure. The use of AA in this study made it possible to compare its results to those obtained with GSH and NAC, which are also acidic agents. AA acidity can modify dentin surfaces through smear layer removal, as observed in our results; this effect might have partially contributed to the observed finding of optimizing the bond strength in addition to its ability to scavenge free radicals. Despite its acidity, AA did not result in significant changes in the microhardness or roughness of H_2_O_2_-treated dentin.

In a recent report using a series of in vitro and in vivo studies and clinical trials, NAC application before commencing bleaching led to decreased pain, sensitivity, and potential damage to tooth structure and oral mucosa [40]. The same study concluded that NAC has the additional benefit of protecting stem cells from the action of bleaching agents. These findings reflect the safety of this agent and hence the choice of its use in our study [40]. In this study, NAC reversed the negative effect of H_2_O_2_ on adhesion at both applications times. However, as with AA, the 1 min application resulted in higher SBS values compared to the 5 min application, and this could also be attributable to deeper dentinal demineralization at the latter application time, due to the acidic nature of NAC. It is noteworthy to mention here that the results of NAC were comparable to AA at both application times. Nonetheless, NAC and AA resulted in higher SBS values compared to GSH at the 1 min application of these antioxidants, but the difference diminished at the 5 min application. These variances might be ascribed to the molecular differences between these agents; the molecular weight of GSH is almost double that of NAC and AA and, hence, the latter two agents might be more effective in scavenging residual ROS through better subsurface penetration within the dentin substrate.

Although STS is regarded as a strong antioxidant and the gold standard agent to reverse the negative oxidizing effect of NaOCl [18], in our study STS resulted in the lowest SBS values compared to all other used antioxidants and the two negative control groups. This reflects the inability of STS to fully restore the capacity of dentin for bonding and its inadequate neutralization of the ROS generated by H_2_O_2_. The effectiveness of STS in restoring bonding to NaOCl-treated dentin, but not to H_2_O_2_-treated dentin, might be attributed to several factors. Firstly, the forms and clinical concentrations of these oxidizing agents are different: in endodontics NaOCl is usually used in a form of solution with concentrations not higher than 5.25%, whereas the most commonly used concentration of H_2_O_2_ is 35%, which comes in the form of a gel that sticks to the surface and is not easily rinsed off. Moreover, the extended application time of H_2_O_2_ during the bleaching procedure might result in a substrate that is more highly oxidized compared to the effect of NaOCl. Furthermore, unlike the other antioxidants used in this study, the inferior ability of STS to reverse the effect of H_2_O_2_ is partially ascribed to its ineffectiveness in modifying the dentin surface through smear layer removal, which could potentially be saturated with free radicals that prevent optimal adhesion.

The effect of the used agents on roughness and microhardness might give an indication of their ability to enhance or lower bonding to dentin. Controlled increases in surface roughness are known to enhance bonding [41]. The effect of these agents on dentin characteristics needs to be studied further but, according to our results, all of the agents did not result in significant changes in roughness or microhardness of H_2_O_2_-treated dentin. When applied for 5 min, GSH yielded a Δ roughness value higher than that of all of the other antioxidants, except for NAC when applied for the same duration. It is also interesting to mention that the 1 min application of GSH or STS resulted in a negative Δ roughness, while their extended application resulted in a positive Δ roughness. The explanation for this observation remains unknown. In a recent study, a 20% concentration of NAC resulted in a marked reduction in root dentin microhardness [42]; nevertheless, this concentration is four times higher than that used in our investigation.

Interest in utilizing NAC in dentistry is expanding due to its properties, which can be harnessed in different applications [23,24]. We believe that the use of NAC within a dental bleaching protocol not only improves bonding but may also deactivate MMPs and protect the pulp cells, which would translate clinically into an improved longevity of resin-based restorations and less post-operative sensitivity. Despite the fact that the therapeutic use of antioxidants is acknowledged in the dental field, since they are likely safe and biocompatible to oral tissues when used appropriately, their clinical application might be limited due to their short shelf life, which necessitates the preparation of a fresh solution for each use [18].

## 5. Conclusions

The limitations of this investigation include, but are not limited to, the use of extracted human teeth from various subjects and the use of in vitro experimental conditions that give only limited answers to more complex problems. Within these limitations, resin SBS to H_2_O_2_-treated dentin was improved by NPSH (GSH and NAC). NAC, however, produced a greater SBS than GSH at 1 min application, but comparable values to those of delayed bonding and the gold-standard antioxidant (AA). The used antioxidants did not cause significant changes in the roughness and microhardness of H_2_O_2_-treated dentin. H_2_O_2_ did not remove the dentinal smear layer while all antioxidants, except STS, were able to effectively remove it at both application times. In addition to their function as free radical scavengers of oxygen, the ability of AA, NAC, and GSH to eliminate the oxidized smear layer is believed to be one of the mechanisms by which they improve bonding to oxidized dentin. The data obtained in this investigation have clinical relevance as the capacity to restore teeth immediately post-bleaching by the application of effective antioxidants prior to the bonding procedure avoids needless discomfort and inconvenience for the patients caused by delayed bonding. However, further in vitro and ex vivo studies and clinical trials on the effect of these agents on orally related cells, the physical characteristics of tooth structure, and overall bleaching efficacy are warranted before this concept can be recommended for clinical use.

## Figures and Tables

**Figure 1 polymers-16-01473-f001:**
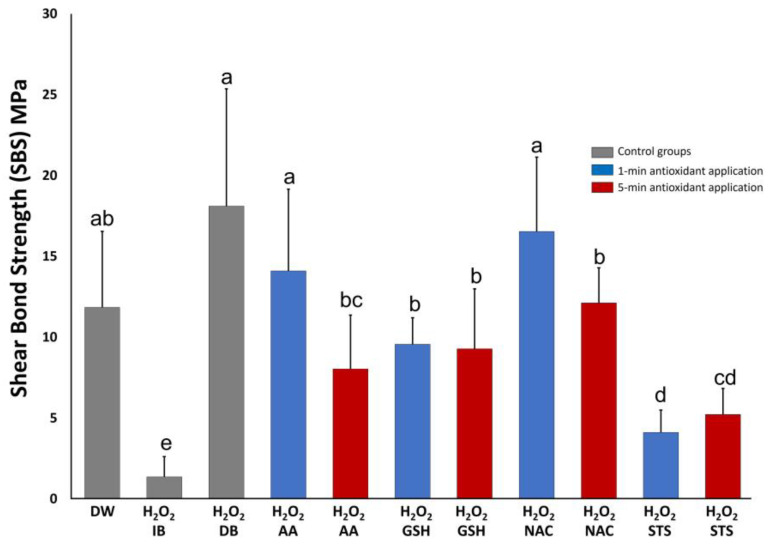
Mean shear bond strength value and standard deviation (MPa ± SD) of resin to dentin of the different experimental and control groups. Groups identified by different lower-case letters indicate significant statistical difference (*p* < 0.05). H_2_O_2_: hydrogen peroxide, AA: ascorbic acid, GSH: reduced glutathione, NAC: *n*-acetylcysteine, STS: sodium thiosulfate, DW: distilled water, IB: immediate bonding, DB: delayed bonding.

**Figure 2 polymers-16-01473-f002:**
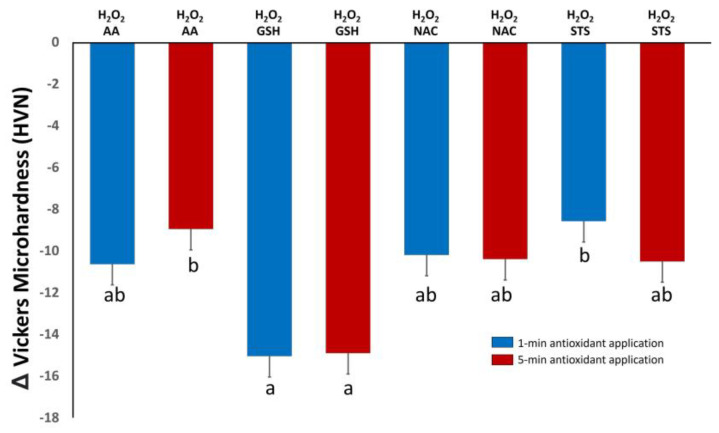
Mean and standard deviation (SD) of dentin Δ microhardness (HVN). Different letters indicate a statistically significant difference at 5%. H_2_O_2_: hydrogen peroxide, AA: ascorbic acid, GSH: reduced glutathione, NAC: *n*-acetylcysteine, STS: sodium thiosulfate.

**Figure 3 polymers-16-01473-f003:**
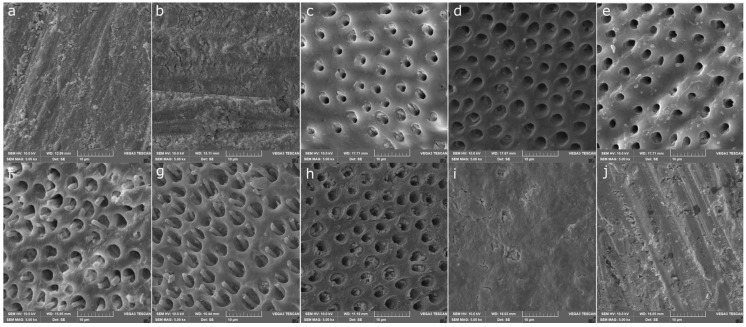
Representative scanning electron microscopy images of dentin topography after each treatment protocol. A smear layer was created on each dentin surface before further treatment of dentin with distilled water, hydrogen peroxide, and/or antioxidants. (**a**) Dentin surface treated with distilled water for 15 min. (**b**) Dentin surface treated with 35% hydrogen peroxide for 15 min. (**c**,**d**) Dentin surface treated with 35% hydrogen peroxide for 15 min followed by 5% ascorbic acid for 1 and 5 min, respectively. (**e**,**f**) Dentin surface treated with 35% hydrogen peroxide for 15 min followed by 5% reduced glutathione for 1 and 5 min; respectively. (**g**,**h**) Dentin surface treated with 35% hydrogen peroxide for 15 min followed by 5% *n*-acetylcysteine for 1 and 5 min, respectively. (**i**,**j**) Dentin surface treated with 35% hydrogen peroxide for 15 min followed by 5% sodium thiosulfate for 1 and 5 min, respectively.

**Table 1 polymers-16-01473-t001:** Protocol of dentin surface conditioning for shear bond strength test. All groups received bonding immediately after conditioning, except group 11 which received bonding 2 weeks after treatment with hydrogen peroxide. H_2_O_2_: hydrogen peroxide, AA: ascorbic acid, GSH: reduced glutathione, NAC: *n*-acetylcysteine, STS: sodium thiosulfate, DW: distilled water.

Group	Dentin Surface Treatment
Group 1	35% H_2_O_2_ (pH 4.5) for 15 min and 5% AA (pH = 2.29) for 1 min
Group 2	35% H_2_O_2_ for 15 min and 5% AA for 5 min
Group 3	35% H_2_O_2_ for 15 min and 5% GSH (pH = 2.88) for 1 min
Group 4	35% H_2_O_2_ for 15 min and 5% GSH for 5 min
Group 5	35% H_2_O_2_ for 15 min and 5% NAC (pH = 1.95) for 1 min
Group 6	35% H_2_O_2_ for 15 min and 5% NAC for 5 min
Group 7	35% H_2_O_2_ for 15 min and 5% STS (pH = 7.24) for 1 min
Group 8	35% H_2_O_2_ for 15 min and 5% STS for 5 min
Group 9	(First negative control): DW (pH 7.03) for 15 min
Group 10	(Positive control): 35% H_2_O_2_ for 15 min
Group 11	(Second negative control): 35% H_2_O_2_ for 15 min (bonding after 2 weeks)

**Table 2 polymers-16-01473-t002:** Protocol of dentin surface conditioning for microhardness and roughness tests. H_2_O_2_: hydrogen peroxide, AA: ascorbic acid, GSH: reduced glutathione, NAC: *n*-acetylcysteine, STS: sodium thiosulfate.

Group	Dentin Surface Treatment of the Sectioned Discs
Group 1	Baseline: H_2_O_2_ 15 min, Treatment: H_2_O_2_ 15 mins and AA for 1 min
Group 2	Baseline: H_2_O_2_ 15 min, Treatment: H_2_O_2_ 15 mins and AA for 5 min
Group 3	Baseline: H_2_O_2_,15 min, Treatment: H_2_O_2_ 15 mins and GSH for 1 min
Group 4	Baseline: H_2_O_2_ 15 min, Treatment: H_2_O_2_ 15 mins and GSH for 5 min
Group 5	Baseline: H_2_O_2_ 15 min, Treatment: H_2_O_2_ 15 mins and NAC for 1 min
Group 6	Baseline: H_2_O_2_ 15 min, Treatment: H_2_O_2_ 15 mins and NAC for 5 min
Group 7	Baseline: H_2_O_2_ 15 min, Treatment: H_2_O_2_ 15 mins and STS for 1 min
Group 8	Baseline: H_2_O_2_ 15 min, Treatment: H_2_O_2_ 15 mins and STS for 5 min

**Table 3 polymers-16-01473-t003:** Mean and standard deviation (SD) of dentin Δ roughness (Sa). Different letters indicate statistically significant difference at 5%. H_2_O_2_: hydrogen peroxide, AA: ascorbic acid, GSH: reduced glutathione, NAC: *n*-acetylcysteine, STS: sodium thiosulfate.

Group	Δ Roughness (Mean ± SD)	
	1 min Antioxidant Application	5 min Antioxidant Application
H_2_O_2_/AA	0.00021 ± 0.00023 ^abc^	0.00042 ± 0.00028 ^bc^
H_2_O_2_/GSH	−0.00012 ± 0.00039 ^ab^	0.0011 ± 0.00073 ^d^
H_2_O_2_/NAC	0.00006 ± 0.00011 ^abc^	0.00057 ± 0.00029 ^cd^
H_2_O_2_/STS	−0.00017 ± 0.00010 ^a^	0.00016 ± 0.00011 ^abc^

## Data Availability

The authors confirm that the data supporting the findings of this study are available within the article and that the raw data are available on request from the corresponding author.

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
