# Peer review of "In Vitro Investigation of the Effects of Various Reducing Agents on Dentin Treated with Hydrogen Peroxide"

_polymers, 2024, doi:10.3390/polym16111473_

Round 1
Reviewer 1 Report
Comments and Suggestions for Authors
The authors presented a good study about Non-Protein Thiols: Reversal Agents for Hydrogen Peroxide-Oxidized Dentin. There are some main comments:
Abstract:
the conclusion section does not support the results. Please improve.
Introduction:
The re are some more recent and relevant references can be added. Besides, more justification is needed for the study.
Materials
Please add the trade name for the materials and devices.
Methods:
I think you need TEM image as well. Please add the images for this technique or explain the reasons for not using it.
The number of tests is not enough for publication in this journal. You need some more tests in this study.
Discussion
The discussion section is so long and confusing. Please improve it. Some data are extra in it.
Author Response
Dear Respected Reviewer
On behalf of the co-authors, we would like to extend our sincere appreciation for the time and efforts spent to review the manuscript. The authors have made changes to the manuscript according to the valuable comments and suggestions. We strongly believe that this paper will be of a great interest to the readers of Polymers and sincerely hope that the manuscript is now suitable for publication in this reputable journal.
Response to reviewer's comments:
Reviewer 1:
Comment 1: Abstract: The conclusion section does not support the results. Please improve.
Response: The conclusion in the abstract has been amended to reflect more on the findings of the study such as the effect of roughness, microhardness and smear layer removal.
Comment 2: Introduction: There are some more recent and relevant references can be added. Besides, more justification is needed for the study.
Response: Further justification and new recent references from 2024 and 2023 have been added to the manuscript. Changes and new references are highlighted in yellow in the text and references list.
Comment 3: Materials: Please add the trade name for the materials and devices.
Response: The trade name of materials and devises are added and highlighted in the text in the methodology section.
Comment 4: Methods: I think you need TEM image as well. Please add the images for this technique or explain the reasons for not using it. The number of tests is not enough for publication in this journal. You need some more tests in this study.
Response: In our investigation four experiments were conducted including: shear bond strength, roughness, microhardness and SEM evaluation of the surface morphology. The main objective revolved around the effect of the used antioxidants on the shear bond strength of bleached dentin and to further explain the results of this objective, it was necessary to study other parameters which have direct effect on the observed findings of the shear bond strength. As mentioned in the discussion, changes in the roughness and/or hardness of the substrate might have an direct impact on bonding, and this impact can be a negative or a positive one. Furthermore, the effect on the smear layer on the dentin surface was reported, as it is known that modification of this layer influences bonding to a great extent. In our study we proved for the first time that antioxidants reverse the negative effect of hydrogen peroxide not only through their neutralizing ability of this oxidizing agent and its by-products by also by inducing changes in the smear layer that is believed to be saturated with the oxidizing products that hamper effective resin-dentin bonding. The reviewer also raised a valid point regarding the potential use of TEM to further investigate the samples. While we acknowledge the merit of this approach, we chose not to employ it in our study due to several considerations: our main concern was the effect on the surface of dentin that would receive the bonding procedure, and this called for the use of SEM to study the surface topography. The use of the latter method also aligns with the majority of reports in the literature on the topic that utilized this technique for the purpose of studying bleached tooth structure. Furthermore, resources constraints and budget limitations were other factors. Given the scope of our study and available resources, we prioritized certain methodologies that were deemed essential for addressing our research objectives effectively. Despite this, we would like to assure the reviewer that our chosen methodologies were carefully selected to maximize the validity and reliability of our results.
Comment 5: Discussion: The discussion section is so long and confusing. Please improve it. Some data are extra in it.
Response:
The discussion starts with the importance of the studied matter and elaborates more on the selection of the substrate and the testing methodology. Followed by listing the main findings that led to the rejection of the null hypothesis. Then we elaborated on discussing the results obtained with H2O2 and the different antioxidant used on H2O2-treated dentin. As can be noticed in the discussion, the effect of these antioxidants on microhardness, roughness, bonding and smear layer were all covered throughout the text and some correlations were speculated and we also elaborated on the possible role of the pH and the molecular weight of the used antioxidant on the perceived findings. With this sequence, we tried our best to in shaping a narrative for the paper to contextualize the findings within the broader scholarly conversation and guiding future research endeavors. However, we agree on the recommendations of reviewers 1 and 3 that there is merit in streamlining the discussion to enhance the clarity and focus of the paper. We have taken the suggestion into account and have made revisions accordingly. The word count of the revised discussion is 1599 compared to the previous number of 2030 which reflects around 22% reduction in this section.
Reviewer 2 Report
Comments and Suggestions for Authors
This is very interesting manuscript. The interduction is well written. Also, the section materials and mathods is also well written, but I think that authors need to explain why the extraction teeth were in formalin not in arteficial saliva. Sections result and discussion are also well written. Minor English spelling mistakes are detected so it would be good to make one more chech in English.
Comments on the Quality of English LanguageMinor editing of English language required.
Author Response
Dear Respected Reviewer
On behalf of the co-authors, we would like to extend our sincere appreciation for the time and efforts spent to review the manuscript. The authors have made changes to the manuscript according to the valuable comments and suggestions. We strongly believe that this paper will be of a great interest to the readers of Polymers and sincerely hope that the manuscript is now suitable for publication in this reputable journal.
------------------------------------
Reviewer 2:
This is very interesting manuscript. The introduction is well written. Also, the section materials and methods is also well written, but I think that authors need to explain why the extraction teeth were in formalin not in artificial saliva. Sections result and discussion are also well written. Minor English spelling mistakes are detected so it would be good to make one more check in English.
Response:
Thank you for feedback on the manuscript. Regarding the storage medium of extracted teeth, besides the needed property of the medium to keep the extracted teeth moist, it is also essential that the medium has the ability to prevent bacterial growth. The antimicrobial properties of the storage medium are crucial aspect, as extracted teeth pose a risk of cross-infection and this property is not found in artificial saliva. Different solutions have been suggested in the literature for this purpose such as thymol, sodium hypochlorite (NaOCl), glutaraldehyde, formalin, etc. However, another consideration that needs to be taken into account especially for the purpose of our study is the effect of the storage medium on bonding of the tooth structure to resin. For instance, NaOCl is known to negatively impact resin-dentin bonding, thus we opt to use formalin. According to Lee at al study in the Journal of American Dental Association, formalin might be the best option for storage of teeth to be used for in vitro dental bonding studies.
Lee JJ, Nettey-Marbell A, Cook A Jr, Pimenta LA, Leonard R, Ritter AV. Using extracted teeth for research: the effect of storage medium and sterilization on dentin bond strengths. J Am Dent Assoc. 2007 Dec;138(12):1599-603. doi: 10.14219/jada.archive.2007.0110. PMID: 18056105.
Regarding minor English corrections, we made the necessary revisions for the English language to ensure clarity and accuracy in the manuscript.
Reviewer 3 Report
Comments and Suggestions for Authors
The manuscript is devoted to a rather interesting topic, however, the data presented in the manuscript can be predicted from general considerations. It has been known for more than half a century that antioxidants prevent or reduce the negative effects of oxidizing agents. It is also known that sulfhydryl compounds have the greatest activity. In this regard, the level of novelty can be assessed as not high.
The name (Non-Protein Thiols: Reversal Agents for Hydrogen Peroxide-Oxidized Dentin) does not correspond to the results obtained by the authors. Firstly, ascorbic acid and sodium thiosulfate are not thiols. Secondly, the article does not indicate any reverse mechanism of action. Hydrogen peroxide and antioxidant are added together! This gives rise to another problem; perhaps antioxidants simply reduce the concentration of hydrogen peroxide by interacting with it. This issue has remained unexplored and requires careful study. The manuscript lacks control experiments almost everywhere. For example, not a single experiment tests the effect of antioxidants themselves on dentin. How does the shear bond strength of dentin change when exposed to NAC in the absence of an oxidizing agent? More experiments need to be done! The authors measure Vickers microhardness dentin, the data are presented in differential form. From experience we know that dentin has 40-50 HVN. If you present the data as real measurement values, then all the difference may disappear. I suggest that the authors rearrange the data in real dimensions rather than in differential form. In Table 3, something is wrong with the values. Apparently zeros are missing. The data is again presented in differential form. It is necessary to provide real values! Electron microscopy data contradict the authors' narrative: control and 35% hydrogen peroxide do not damage the surface layer, but in the presence of antioxidants, damage is observed. This is another reason to make dentin controls with antioxidants! Overall, the manuscript needs detailed revision and shortening of the Discussion section.
Author Response
Dear Respected Reviewer
On behalf of the co-authors, we would like to extend our sincere appreciation for the time and efforts spent to review the manuscript. The authors have made changes to the manuscript according to the valuable comments and suggestions. We strongly believe that this paper will be of a great interest to the readers of Polymers and sincerely hope that the manuscript is now suitable for publication in this reputable journal.
Reviewer 3:
The manuscript is devoted to a rather interesting topic, however, the data presented in the manuscript can be predicted from general considerations. It has been known for more than half a century that antioxidants prevent or reduce the negative effects of oxidizing agents. It is also known that sulfhydryl compounds have the greatest activity. In this regard, the level of novelty can be assessed as not high. The name (Non-Protein Thiols: Reversal Agents for Hydrogen Peroxide-Oxidized Dentin) does not correspond to the results obtained by the authors. Firstly, ascorbic acid and sodium thiosulfate are not thiols. Secondly, the article does not indicate any reverse mechanism of action. Hydrogen peroxide and antioxidant are added together! This gives rise to another problem; perhaps antioxidants simply reduce the concentration of hydrogen peroxide by interacting with it. This issue has remained unexplored and requires careful study. The manuscript lacks control experiments almost everywhere. For example, not a single experiment tests the effect of antioxidants themselves on dentin. How does the shear bond strength of dentin change when exposed to NAC in the absence of an oxidizing agent? More experiments need to be done! The authors measure Vickers microhardness dentin, the data are presented in differential form. From experience we know that dentin has 40-50 HVN. If you present the data as real measurement values, then all the difference may disappear. I suggest that the authors rearrange the data in real dimensions rather than in differential form. In Table 3, something is wrong with the values. Apparently zeros are missing. The data is again presented in differential form. It is necessary to provide real values! Electron microscopy data contradict the authors' narrative: control and 35% hydrogen peroxide do not damage the surface layer, but in the presence of antioxidants, damage is observed. This is another reason to make dentin controls with antioxidants! Overall, the manuscript needs detailed revision and shortening of the Discussion section.
Response:
- Regarding novelty and title: The idea of using non-protein thiols (NPSH) in our investigation stems from the fact that these compounds namely reduced glutathione (GSH) and n-acetylcysteine (NAC) were found in our previous studies to prevent toxicity on dental pulp cells and reduce dentinal matrix metalloproteinases (MMPs) activity. The concept of using these agents as reversal agents on bleached dentin has not been explored well in the literature. In our study, the main objective was studying their efficacy in this regard and compare it to conventional or gold standard agents namely ascorbic acid and sodium thiosulfate used in dentistry to neutralize oxidizing compounds.
- In our study, the application method of the used antioxidant did not include mixing them with hydrogen peroxide. Clinically, mixing these agents together before bleaching would abolish the ability of hydrogen peroxide to whiten tooth structure. Thus, the suggested protocol is to add the antioxidant after teeth whitening. In our investigation, as mentioned in the methodology section, on page three “The experimental groups received treatment with 35% H2O2 for 15 minutes followed by rinsing with DW and then conditioning with 5% of either AA, GSH, NAC, STS for 1 or 5 minutes”. We hope this statement clarify the confusion about the application protocol.
- Regarding the control groups, in the shear bond strength testing we had three control groups namely: two negative control groups and one positive control group. The first negative control group is using distilled water for the treatment, and in the second negative control group hydrogen peroxide was used with delayed bonding after 2 weeks. The positive control group received treatment with hydrogen peroxide and immediate bonding. In regard to the roughness and microhardness tests, we mainly focused on the effect of the used antioxidants on these measurements on dentin that is treated with H2O2 and how these effects might be correlated with the bonding results to this substrate. Thus, each sample for these tests were divided into two sections; one for the baseline treatment (H2O2) and the other for the experimental part (antioxidant treatment after H2O2 application), and then the microhardness and roughness values were tested for each section followed by calculating the difference between the values of the two sections from the same sample. Hence, there is no control group showing in the figure or table of the microhardness and roughness tests; respectively, and that’s why the values are shown as delta (Δ).
- For smear layer removal, this is not considered as damage; actually modification or removal of this layer might be of a benefit in the case of bleached dentin as this layer could potentially be saturated with free radicals that prevent optimal adhesion. Thus, for instance, STS was not able to remove the smear layer and its shear bond strength values were the least compared to other types of antioxidants which were effective in removing this layer. In the context of our study, we do not refer to the removal of thus layer as damage, damage might happen in form of erosion of dentin in case the application time of these acidic agents were extended for a long period of time where the underneath sound dentin becomes eroded, and this is not the case of our study.
- The discussion starts with the importance of the studied matter and elaborates more on the selection of the substrate and the testing methodology. Followed by listing the main findings that led to the rejection of the null hypothesis. Then we elaborated on discussing the results obtained with H2O2 and the different antioxidant used on H2O2-treated dentin. As can be noticed in the discussion, the effect of these antioxidants on microhardness, roughness, bonding and smear layer were all covered throughout the text and some correlations were speculated and we also elaborated on the possible role of the pH and the molecular weight of the used antioxidant on the perceived findings. With this sequence, we tried our best to in shaping a narrative for the paper to contextualize the findings within the broader scholarly conversation and guiding future research endeavors. However, we agree on the recommendations of reviewers 1 and 3 that there is merit in streamlining the discussion to enhance the clarity and focus of the paper. We have taken the suggestion into account and have made revisions accordingly. The word count of the revised discussion is 1599 compared to the previous number of 2030 which reflects around 22% reduction in this section.
Round 2
Reviewer 1 Report
Comments and Suggestions for Authors
It can be accepted.
Author Response
Dear reviewer
Thank you for the recommendation to accept the paper. We appreciate the time and efforts spent to review the manuscript.
Sincerely
Corresponding author
Reviewer 3 Report
Comments and Suggestions for Authors
Dear Authors, please respond to my comments. Currently you have responded to some other, albeit related, comments. I have posted my comments below for your convenience.
The manuscript is devoted to a rather interesting topic, however, the data presented in the manuscript can be predicted from general considerations. It has been known for more than half a century that antioxidants prevent or reduce the negative effects of oxidizing agents. It is also known that sulfhydryl compounds have the greatest activity. In this regard, the level of novelty can be assessed as not high.
The name (Non-Protein Thiols: Reversal Agents for Hydrogen Peroxide-Oxidized Dentin) does not correspond to the results obtained by the authors. Firstly, ascorbic acid and sodium thiosulfate are not thiols. Secondly, the article does not indicate any reverse mechanism of action. Hydrogen peroxide and antioxidant are added together! This gives rise to another problem; perhaps antioxidants simply reduce the concentration of hydrogen peroxide by interacting with it. This issue has remained unexplored and requires careful study. The manuscript lacks control experiments almost everywhere. For example, not a single experiment tests the effect of antioxidants themselves on dentin. How does the shear bond strength of dentin change when exposed to NAC in the absence of an oxidizing agent? More experiments need to be done! The authors measure Vickers microhardness dentin, the data are presented in differential form. From experience we know that dentin has 40-50 HVN. If you present the data as real measurement values, then all the difference may disappear. I suggest that the authors rearrange the data in real dimensions rather than in differential form. In Table 3, something is wrong with the values. Apparently zeros are missing. The data is again presented in differential form. It is necessary to provide real values! Electron microscopy data contradict the authors' narrative: control and 35% hydrogen peroxide do not damage the surface layer, but in the presence of antioxidants, damage is observed. This is another reason to make dentin controls with antioxidants! Overall, the manuscript needs detailed revision and shortening of the Discussion section.
Author Response
Dear Reviewer
Thank you again for the time and efforts spent on reviewing our manuscript. Please find below a point-by-point response to the valuable and insightful raised comments.
Comment 1 (Novelty and title comment): The manuscript is devoted to a rather interesting topic, however, the data presented in the manuscript can be predicted from general considerations. It has been known for more than half a century that antioxidants prevent or reduce the negative effects of oxidizing agents. It is also known that sulfhydryl compounds have the greatest activity. In this regard, the level of novelty can be assessed as not high. The name (Non-Protein Thiols: Reversal Agents for Hydrogen Peroxide-Oxidized Dentin) does not correspond to the results obtained by the authors. Firstly, ascorbic acid and sodium thiosulfate are not thiols.
Response to comment 1: Regarding the novelty and title: The idea of using non-protein thiols (NPSH) in our investigation stems from the fact that these compounds namely reduced glutathione (GSH) and n-acetylcysteine (NAC) were found in our previous studies to prevent toxicity on dental pulp cells and reduce dentinal matrix metalloproteinases (MMPs) activity. However, the concept of using these agents as reversal agents on bleached dentin is poorly explored in the literature and thus it warrants further studies, hence conducting the current investigation. In our study, the main objective was studying the effect of GSH and NAC on adhesion, roughness, microhardness, and topography of H2O2-treated dentin and compare the results to conventional or gold standard agents which are ascorbic acid and sodium thiosulfate that are used in clinical dentistry and dental research to neutralize oxidizing compounds. However, based on the recommendation of the reviewer, we opted to change the title to “In vitro investigation of the effects of various reducing agents on dentin treated with hydrogen peroxide”.
Comment 2 (Mixing H2O2 with the antioxidants): The article does not indicate any reverse mechanism of action. Hydrogen peroxide and antioxidant are added together! This gives rise to another problem; perhaps antioxidants simply reduce the concentration of hydrogen peroxide by interacting with it. This issue has remained unexplored and requires careful study.
Response to comment 2: In our study, the application method of the used antioxidant did not include mixing them with H2O2. Actually, mixing these agents with H2O2 before the bleaching procedure would abolish the ability of H2O2 to whiten tooth structure. Thus, the suggested clinical protocol is to apply the antioxidant on tooth structure after teeth whitening. As mentioned in the methodology section of our investigation (page 3): “The experimental groups received treatment with 35% H2O2 for 15 minutes followed by rinsing with DW and then conditioning with 5% of either AA, GSH, NAC, STS for 1 or 5 minutes”. We hope this statement clarify the confusion about the application protocol.
Comment 3 (Control groups): The manuscript lacks control experiments almost everywhere. For example, not a single experiment tests the effect of antioxidants themselves on dentin. How does the shear bond strength of dentin change when exposed to NAC in the absence of an oxidizing agent? More experiments need to be done!
Response to comment 3: Regarding the control groups, in the shear bond strength testing we had three control groups namely: two negative control groups and one positive control group. The first negative control group is using distilled water for the treatment, and in the second negative control group H2O2 was used with delayed bonding after 2 weeks. The positive control group received treatment with H2O2 with immediate bonding. Regarding the roughness and microhardness tests, we mainly focused on the effect of the used antioxidants on these parameters of H2O2-treated dentin to interpret how these effects might be correlated with the results of resin bond strength to this oxidized substrate. Thus, each sample (disc) used for these tests was divided into two sections (two halves); one half for the baseline treatment (H2O2 application) [R1] and the other half for the experimental part (antioxidant treatment after H2O2 application) [R2], and then the microhardness and roughness were tested for the surface of each half, followed by calculating the difference in the obtained values of the two halves coming from the same sample (disc) [Δ=R2-R1]. Hence, there is no control group per se in the figure or table of the microhardness and roughness tests, and that is why the values are shown as delta (Δ). Each calculation of the Δ microhardness and Δ roughness took into consideration the value of the baseline treatment of each sample which can be regarded as the control.
Concerning the effect of the antioxidants on dentin without pretreatment with H2O2, the idea itself seems of a value. However, these antioxidants are suggested for clinical use after exposure of dentin to oxidizing compounds to reverse oxidized dentin to a reduced state that is more receptive to resin-dentin bonding. However, we previously conducted a study on the effect of GSH on bond strength to non-oxidized dentin (no prior H2O2 application) and the results showed that GSH had neither positive nor negative influence on bond strength (1). However, a long-term positive effect of NPSH on bonding might be observed on non-oxidized dentin as we previously reported that NPSH have a role in deactivating dentinal MMPs which play a major role in degrading resin-dentin bond strength on the long-term (2). Our future research ideas aim to delve deeper into understanding the long-term effects of various reducing agents on oxidized dentin and explore additional parameters taking into account the insights that we mentioned in the discussion on the role of H2O2 in activating dentinal MMPs and the inhibitory effect of NPSH on these proteinases.
- Nassar M, Hiraishi N, Islam MS, Tamura Y, Otsuki M, Kasugai S, Ohya K, Tagami J, Tay FR. The effect of glutathione on 2-hydroxyethylmethacrylate cytotoxicity and on resin-dentine bond strength. Int Endod J. 2014 Jul;47(7):652-8.
- Nassar M, Hiraishi N, Shimokawa H, Tamura Y, Otsuki M, Kasugai S, Ohya K, Tagami J. The inhibition effect of non-protein thiols on dentinal matrix metalloproteinase activity and HEMA cytotoxicity. J Dent. 2014 Mar;42(3):312-8
Comment 4: (VHN values and missing zeros in Table 3): The authors measure Vickers microhardness dentin, the data are presented in differential form. From experience we know that dentin has 40-50 HVN. If you present the data as real measurement values, then all the difference may disappear. I suggest that the authors rearrange the data in real dimensions rather than in differential form. In Table 3, something is wrong with the values. Apparently, zeros are missing. The data is again presented in differential form. It is necessary to provide real values.
Response to comment 4: Thanks for the valuable comment. As described in our response to comment 3, regarding the roughness and microhardness tests, we mainly focused on the effect of the used antioxidants on these parameters of H2O2-treated dentin to interpret how these effects might be correlated with the results of resin bond strength to this oxidized substrate. Thus, each sample (disc) used for these tests was divided into two sections (two halves); one half for the baseline treatment (H2O2 application) [R1] and the other half for the experimental part (antioxidant treatment after H2O2 application) [R2], and then the microhardness and roughness were tested for the surface of each half, followed by calculating the difference in the obtained values of the two halves coming from the same sample (disc) [Δ=R2-R1]. Hence the numerical value for Δ depends on how much difference the antioxidant caused on the microhardness or roughness of H2O2-treated dentin. We believe that this method of presenting the data helps the reader comprehend the effect that each antioxidant has on the oxidized substrate, whether a negative or positive influence on each measured parameter.
The zeros before the decimal in Table 3 have been added to the values of the standard deviations.
Comment 5 (Smear layer damage): Electron microscopy data contradict the authors' narrative: control and 35% hydrogen peroxide do not damage the surface layer, but in the presence of antioxidants, damage is observed. This is another reason to make dentin controls with antioxidants.
Response to comment 5: Regarding smear layer removal, removal of this layer is not considered as damage. Actually, modification or removal of this layer might be of a benefit in the case of bleached dentin as this layer could potentially be saturated with free radicals that prevent optimal resin adhesion to dentin. For instance, STS was not able to remove the smear layer and at the same time the shear bond strength value of its group was the least compared to other types of antioxidants which were effective in removing this layer. In the context of our study, we do not refer to the removal of thus layer as damage, damage might happen in form of erosion of dentin in case the application time of these acidic agents were extended for a long period of time where the underneath sound dentin becomes eroded, and this is not the case of our study. On a side note, this layer is also recommended to be removed in other clinical situations such as root canal treatment and certain adhesive procedures.
Comment 6 (Lengthy discussion): Overall, the manuscript needs detailed revision and shortening of the Discussion section.
Response to comment 6: The discussion starts with the importance of the studied matter and elaborates more on the selection of the substrate and the testing methodology. Followed by listing the main findings that led to the rejection of the null hypothesis. Then we elaborated on discussing the results obtained with H2O2 and the different antioxidant used on H2O2-treated dentin. As can be noticed in the discussion, the effect of these antioxidants on microhardness, roughness, bonding and smear layer were all covered throughout the text and some correlations were speculated and we also elaborated on the possible role of the pH and the molecular weight of the used antioxidant on the perceived findings. With this sequence, we tried our best to in shaping a narrative for the paper to contextualize the findings within the broader scholarly conversation and guiding future research endeavors. However, we agree on the recommendations of reviewers 1 and 3 that there is merit in streamlining the discussion to enhance the clarity and focus of the paper. We have taken the suggestion into account and have made revisions accordingly. The word count of the revised discussion is 1599 compared to the previous number of 2030 which reflects around 22% reduction in this section.
Thank you again for the valuable comments and insights and we hope that you find our manuscript suitable for publication in Polymers.
Sincerely
Corresponding author
9th of May 2024

Round 3
Reviewer 3 Report
Comments and Suggestions for Authors
Dear authors! I'm glad you corrected the typos and incorrect numbers. The manuscript still lacks control experiments; the authors’ reasoning that the authors once did these experiments does not convince me that the authors are right. If the authors have data from control experiments, then these data should be presented in the manuscript. If this data is not available, then the authors’ statements are highly questionable. The authors present the data in a differential form - this is not acceptable for serious scientific research. I encourage the authors to present the data in a generally accepted form and then discuss the results obtained. The microscopy data is in conflict with other data. The authors' reasoning is generally understandable, but the data obtained in the manuscript is questionable. I assume that when the necessary controls are made and the data are presented in a generally accepted form, this problem will also be resolved
Author Response
Dear Reviewer
Thank you for your response to the second review. We believe there is a misunderstanding regarding the concept of having the values in delta format. All baseline treatments of one half of the discs and the experimental half of the discs were conducted with only hydrogen peroxide, followed by antioxidant treatment for only the experimental halves, and the difference was used to calculate the hardness and roughness values in delta format. All raw data will be made available by the corresponding author. However, including them in the text will cause uncertainty for the reader as to how to interpret the results. Though we think your comment is valuable, we believe that the way the data is currently presented makes it easier to correlate bond strength data with roughness and hardness testing results. As for the SEM images, we have defended our findings in the rebuttal letter and also explained things related to that matter in the manuscript. We do not understand where the confusion is coming from; as the results of SEM well explain the bond strength values in a way that make it clear that the ability of a certain antioxidant to reverse the effect of hydrogen peroxide on dentin comes from not only its reducing action but also the smear layer removal capacity.
Thank you again and we welcome any decision from your side based on our current and previous replies.
Sincerely
Corresponding author
Round 4
Reviewer 3 Report
Comments and Suggestions for Authors
Dear authors. I'm glad that many aspects of the manuscript have improved. MY main concern is the lack of result in the manuscript. The concern is based on the inability to evaluate primary data. I suggest that authors add their primary data as an appendix to the manuscript to avoid possible criticism from the scientific community.
Author Response
On behalf of the co-authors, we would like to extend our sincere appreciation for the tremendous time and efforts spent to review the manuscript. We have made changes according to the recommended minor revision in which we added the suggested data of the roughness and microhardness tests for each baseline treatment and the related groups. The two new figures can be found in the supplementary material as Figure S1 and S2, and they referred to in the manuscript on page 7.